# Relationships between perceived measures of internal load and wellness status during overseas futsal training camps

Yung-Sheng Chen[1,2,3], Filipe Manuel Clemente[4,5], Jeffrey Cayaban Pagaduan[6], Zachary J. Crowley-McHattan[7], Yu-Xian Lu[1,8], Chia-Hua Chien[2], Pedro Bezerra[4,9], Yi-Wen Chiu[10], Cheng-Deng Kuo[3,11]*

**1** Department of Exercise and Health Sciences, University of Taipei, Taipei, Taiwan, **2** Exercise and Health Promotion Association, New Taipei City, Taiwan, **3** Tanyu Research Laboratory, Taipei, Taiwan, **4** Escola Superior Desporto e Lazer, Instituto Politécnico de Viana do Castelo, Viana do Castelo, Portugal, **5** Instituto de Telecomunicações, Delegação da Covilhã, Lisboa, Portugal, **6** Institute of Active Lifestlye, Palacký University, Olomouc, Czechia, **7** Discipline of Sport and Exercise Science, Faculty of Health, Southern Cross University, Lismore, Australia, **8** Graduate Institute of Athletes and Coaching Science, National Taiwan Sport University, Taoyuan, Taiwan, **9** The Research Centre in Sports Sciences, Health Sciences and Human Development, Vila Real, Portugal, **10** Department of Physical Education, Fu Jen Catholic University, New Taipei City, Taiwan, **11** Department of Medical Research, Taipei Veterans General Hospital, Taipei, Taiwan

* cdkuo23@gmail.com

**Data Availability Statement:** All relevant data are within the paper and its Supporting Information files.

## Abstract

Exercise and sport practitioners frequently utilize rating of perceived exertion (RPE) to evaluate the players' psychophysiological strains during training sessions. The subjective rating of physical exertion level during sports training has been shown to have a reciprocal relationship with wellness status during periodic training or competitive seasons. However, the relationship between subjective physical exertions and wellness status during short-term overseas training camps (OTCs) has not been extensively investigated. This study aimed to examine the perceived responses of physical exertions [session-RPE (sRPE), training monotony, and training strain] and wellness status (fatigue, sleep, delayed onset muscle soreness, stress, and mood) measures in elite young adult futsal players from four separate OTCs with different training tasks. Twenty-seven U-20 male national team futsal players voluntarily participated in this study. The players recruited for OTCs were based on their performance during domestic training camps and the tactical demand of the team. The task of each OTCs was defined as: 1) 1st OTC = game-based camp (n = 14); 2) 2nd and 3rd OTC = training-based camp (n = 20 and n = 17, respectively); and 3) 4th OTC = pre-tournament camp (n = 14). The OTCs consisted of 11 training sessions (18.9 hours) and 16 friendly matches (23.8 hours). During daily training sessions and friendly matches, sRPE was used to quantify training load (TL). Additionally, a five-elements general wellness questionnaire was used to evaluate daily wellness status in the morning. The results demonstrated that the mean and sum sRPE in the game-based OTCs were significantly lower compared to the mean sRPE [$p < 0.01$, effect size (ES) = -4.8; $p < 0.01$, ES = -2.9] and sum sRPE in the training-based OTCs ($p < 0.01$, ES = -3.6; $p < 0.01$, ES = -3.1). The mean ($p = 0.01$; ES = -2.0) and sum sRPE ($p < 0.01$; ES = -3.4) in the game-based OTC were also lower than that in the pre-tournament OTC. Conversely, the wellness scores in the game-based OTC were

**Funding:** This work was supported by a grant VGHUST96-P1-06 from VGHUST Joint Research Program, and a grant MOST-103-2410-H-075-001 from the Ministry of Science and Technology, Taiwan. No additional external funding was received for this study.

**Competing interests:** The authors have declared that no competing interests exist.

higher compared to the training-based ($p$ = 0.01; ES = 1.8) and the pre-tournament OTCs ($p$ < 0.01; ES = 1.6). There was a negative relationship between mean and sum sRPE and all wellness scores (mean sRPE = $r$ = -0.441 ~ -0.575, $p$ < 0.001; sum sRPE = $r$ = -0.41 ~ -0.559, $p$ < 0.001). Our findings suggested that responses to training sessions, derived from mean and sum sRPE and wellness scores, are dependent upon the task-specific nature of OTCs among elite futsal players. Utilization of mean and sum sRPE and wellness measures to monitor the psychophysiological health during short-term OTCs is recommended.

## Introduction

Futsal is a variant sport of soccer played indoors on either a wood or artificial floor surface within an area of 38–42 m (length) and 20–25 m (width). The game of futsal requires four field players and one goalkeeper on each team with no limit on the number of substitutions. The typical characteristics of futsal include vigorous game intensity, rapid decision-making, high physiological loads, psychological stress, and heightened emotional states [1, 2]. For example, Sarmento et al. [3] reported that the most common tactical actions to score in the Spanish Primera futsal league were the defense-to-offense transitions and positional play, which are characteristic of heavy workload through short-distance sprints and continuous running patterns during offensive plays. Furthermore, the locomotor demand during the futsal match consists of intermittent high frequency running, rapid change of direction, acceleration, and deceleration in field playing positions [4].

Integrating training and match load monitoring throughout a season is important for optimizing performance outcomes [5]. In professional futsal players, tactical demand, match fixtures, and ranking strategy are essential considerations for appropriate periodization during a season [6]. Conversely, identifying players' competency via short-term training periods during training camps (TCs) is an essential process to ensure the readiness of national teams prior to major tournaments [7]. This period is a crucial stage for helping coaches and strength and conditioning practitioners tabulate the upcoming training program to optimize individual and team performance. One method commonly used to evaluate the strengths and weaknesses of a team's performance during pre-tournament preparation is to conduct overseas training camps (OTCs) [8–10]. Furthermore, engagement in OTCs increases motivation and mental toughness from exposure to different opponents playing styles that are unavailable in domestic competitions [11]. However, most importantly, more matches during a congested international tournament may also contribute additional psychological and physiological strain, requiring players to develop greater resilience. Thus, OTCs provide opportunities for the players to be familiar with daily routines and psychophysiological demands typical of a tournament schedule. Through comprehensive monitoring of cycle changes in players (such as external load, internal load, wellness status, and readiness for training and match), coaches and practitioners can implement appropriate training programmes and recovery strategies in optimizing the team performance during the camps [12].

Quantification of perceived exertion in response to exercise engagement is one popular method of evaluating physiological strain during training/competition. To date, Borg CR-10 is extensively used to subjectively measure physical exertion during sports training due to its simplicity and accessibility [13]. Subsequently, the Borg CR-10 is used throughout a training bout to quantify the training experience regarding training intensity and duration, known as

session-RPE (sRPE) [14]. The validity and reliability of the sRPE in sports training studies has been systematically reviewed by Haddad et al. [15].

Apart from training monitoring, utilizing assessments of stress, recovery, and sleep are also essential factors in understanding players' health condition and wellness status [16, 17]. One common tool to evaluate these variables is the general wellness questionnaire modified by Hopper index [18, 19]. The validity and reliability of the wellness assessments during sports training has been demonstrated in both individual [20] and team sports [21] settings.

The intensity of TL and wellness status depends on the types of training and the objectives of the session [22]. The paradoxical relationship between internal load and wellness status has been reported throughout a season in both soccer [23, 24] and futsal players [8, 16]. We recently demonstrated a moderate effect size (ES) and negative correlation between external/internal TL and muscular perceptions [i.e. delayed onset muscle soreness (DOMS) and fatigue] during a 7-day short-term domestic TC in elite U-20 futsal players [16]. These findings seem to be workload-dependent. Conversely, Chen et al. [8] reported a poor relationship between TL and wellness scores during the first period of domestic TCs (consisting of physical and fitness examination, performance evaluation, etc.) and an international tournament in futsal players. However, the TL and wellness scores during domestic TCs, with a high accumulation of TL and invitational tournaments with congested schedules, were negatively correlated. Additionally, one of our previous reports demonstrated that a 5-day short-term OTC with game-based tasks had no negative impact on wellness status in senior futsal national team players [11].

In light of the abovementioned studies, there seems to be a scarcity of literature regarding TL and wellness status during futsal OTCs. Such information may help coaches and sports practitioners in proper decision-making for training and recovery strategies, even competition management. Consequently, coaches can focus the game tasks and match analysis in relation to individual and collective performance when these pre-match preparations are well organized [25, 26]. Therefore, the purpose of this study was twofold. Firstly, to identify characteristics of the perceived effect of training exertion and wellness status in elite young adult futsal players during different training tasks in OTCs. Secondly, the study aimed to examine the relationship between perceived responses of exercise engagement and the wellness status during OTCs. It was hypothesized that perceived exertion level during training and wellness status would vary from camp to camp. The secondary hypothesis was that there would be a negative relationship between all wellness indices and subjective physical exertion during exercise in OTCs.

## Materials and methods

### Experimental approach to the problem

This study was an observational study that aimed to examine the characteristics of perceived measures of internal TL and wellness status. Daily subjective measures of sRPE and general wellness questionnaire were implemented in four separate OTCs during pre-tournament preparation of a bi-annual continental tournament between July 2018- June 2019. The duration and number of players who participated in the OTC were: 1) 1st OTC (game-based task): 6 days, 14 players (July 28th–August 2nd 2018); 2) 2nd OTC (training-based task): 5 days, 20 players (November 19th–23rd 2018); 3) 3rd OTC (training-based task): 6 days, 17 players (April 7th–12th 2019); 4) 4th OTC (pre-tournament task): 10 days, 14 players (June 1st–10th 2019). Overall, the OTCs consisted of 11 training sessions (18.9 hours) and 16 friendly matches (23.8 hours). Over the data collection period, training load and wellness status were monitored daily. The training schedule is presented in Table 1.

**Table 1. The training schedule of the overseas training camps.**

| Training camps | Day 1 | Day 2 | Day 3 | Day 4 | Day 5 | Day 6 | Day 7 | Day 8 | Day 9 | Day 10 |
|---|---|---|---|---|---|---|---|---|---|---|
| 1st Camp (n = 14) | Travelling | Match | Match | Match | Match | | | | | |
| Game-based camp | Training | | | | | | | | | |
| July 28th–August 2nd 2018 | | | | | | | | | | |
| 2nd Camp (n = 20) | Travelling | Friendly Match | Friendly Match | Training | Friendly Match Travelling | | | | | |
| Training-based camp | | | | | | | | | | |
| November 19th–23rd 2018 | | | | | | | | | | |
| 3rd Camp (n = 17) | Travelling | Friendly Match Training | Training | Friendly Match Training | Training | Friendly Match Travelling | | | | |
| Training-based camp | Training | | | | | | | | | |
| April 7th–12th 2019 | Friendly Match | | | | | | | | | |
| 4th Camp (n = 14) | Travelling | Training | Friendly Match | Training | Friendly Match | Training | Friendly Match | Rest | Friendly Match | Travelling |
| Pre-tournament camp | Friendly Match | | | | | | | | | |
| June 1st–10th 2019 | | | | | | | | | | |

## Participants

Twenty-seven (twenty-four outfield players and three goalkeepers) male futsal players voluntarily participated this study (age = 17.93 ± 0.87 yrs; height = 1.71 ± 0.07 m; body weight = 65.39 ± 9.39 kg; body fat = 12.54 ± 2.76%; maximal aerobic capacity = 51.98 ± 3.07 ml.kg$^{-1}$.min$^{-1}$). Player recruitment for the OTCs was based on their performance during domestic TC and the tactical demand of the team. In this study, the number of players who participated in the OTCs was ten players for all camps, one player for 3 camps, seven players for 2 camps, and nine players for 1 camp. Eligibility criteria for participation in this study were that the players did not miss more than two consecutive training sessions during the study period. The players signed written informed consent forms at the point of recruitment and were all familiarized with testing procedures. This study was approved by the Institution Board of Human Ethics Committee (UT-IRB-2018-068) and undertaken in accordance with the Declaration of Helsinki and its later amendments in 2013.

**Design and procedure.** The experimental procedure is described in our recent studies [8, 9]. The players stayed in a domestic hotel close to an airport one night before international travel. The individual RPE and general wellness questionnaire data were collected and assessed after traveling. For TL monitoring, the players reported their RPE scores to the team sports trainer face to face within 30 min after a training session and within 1-h after the friendly matches [27]. All players reported their wellness scores prior to breakfast. Individual RPE and general wellness scores were recorded in a customized excel spreadsheet via an iPad tablet computer (iPad Pro 9.7 a1673, Apple, CA, USA). The average of individual values in each training camp was used for subsequent data analysis. Collectively, 378 measures were recorded in this study.

**Rating of perceived exertion.** The subjective perceived exertion in training sessions and matches were assessed using the Borg CR10 scale (0 = nothing at all; 10 = extremely strong, almost maximum) [13]. The players were all familiarized with RPE assessment during their regular training sessions. After each training session, the team sports trainer asked the players, "how hard was your training session?" before the players reported an individual RPE score. The RPE score was then multiplied by the training session/match duration for sPRE (arbitrary

units, a.u.) [14]. Furthermore, training monotony (mean of TL divided by its standard deviation) and training strain (sum of TL multiplied by training monotony during a single training camp) was calculated [15, 28, 29]. Testing procedures have been reported in our previous investigations [8, 16].

**General wellness questionnaire.** The general wellness questionnaire was used to assess the daily wellness conditions of players in this study. The questionnaire consists of cognitive perceptions of fatigue, sleep, DOMS, stress, and mood and requires players to answer via a five-point Likert Scale (1 –worst quality to 5 –best quality). The team's sports trainer asked the players, "how do you feel about the level of fatigue status, sleep quality, muscle soreness, mental stress, and mood?" Afterward, the players reported the scores of each item individually. The sum of items, ranging from 5 to 25 points, was used to evaluate fatigue and wellness status [18]. The players reported their scores individually to avoid peer influence.

### Data collection

Daily sRPE was collected for 5, 4, 5, and 8 days during the 1st, 2nd, 3rd, and 4th OTCs, respectively. Additionally, the wellness questionnaire was recorded for 5, 4, 6, and 9 days during the 1st, 2nd, 3rd, and 4th OTC, respectively.

### Statistical analyses

Descriptive data of all variables were calculated as mean and standard deviation (SD). The average and coefficient of variation of measuring variables were used for comparisons. The normality of variables was examined with the Kolmogorov-Smirnov test. As there were different players who participated in OTCs in this study, inter-differences of TL and wellness scores to standard values were used for comparisons among the OTCs. Additionally, one-way analysis of variance (ANOVA) with Bonferroni adjustment was used to compare the variables across the OTCs. The qualitative magnitude was analyzed using Cohen's d ES. The level of ES was interpreted as trivial (0.0–0.2), small (0.2–0.6), moderate (0.6–1.2), large (1.2–2.0), or very large ($> 2.0$) [30]. The relationships between TL variables and wellness scores were assessed by linear regression analysis. Statistical significance was set at $p < .05$ by using SPSS® Statistics version 25.0 (IBM, Armonk, NY, USA) and Microsoft Excel 2016 (Microsoft Corporation, Redmond, WA, USA).

## Results

### Rating of perceived exertion

One of the players failed to report wellness scores in the 1st OTC, leading to exclusion for statistical analysis. In Table 2, the results revealed significant differences in mean sRPE [$F(3, 64)$ = 35.7, $p < 0.01$], sum sRPE [$F(3, 64) = 36.6$, $p < 0.01$], and training monotony [$F(3, 64) =$ 10.5, $p < 0.01$] among the OTCs. The lowest mean and sum sRPE was found in the 1st OTC (335.4 ± 44.9 a.u.; 1676.8 ± 224.4 a.u.), whereas the highest mean and sum sRPE's were found in the 3rd OTC (894.6 ± 247.6 a.u.) and 4th OTC (4509.8 ± 1136 a.u.), respectively. Very large ES of mean and sum sRPE was found when the 1st OTC was compared to the other OTCs. Furthermore, the lowest value of training monotony was found in the 2nd OTC (2.7 ± 0.4 a.u.), whereas the highest training monotony value was found in the 4th OTC (7.4 ± 3.1 a.u.). Very large ES was found between the 1st OTC and the 2nd OTC. In terms of training strain, the lowest value was found in the 2nd OTC (8932.5 ± 2096.2 a.u.), whereas the highest value was found in the 3rd OTC (14731.9 ± 15508.1 a.u.).

**Table 2. Means of internal loads (session rating of perceived exertion, training monotony, and training strain) and wellness status (wellness, fatigue, delayed onset muscle soreness, sleep, stress, and mood) in four overseas training camps.**

| | Overseas training camps | | | | P value (Effect size) | | | | | |
|---|---|---|---|---|---|---|---|---|---|---|
| | 1st camp (n = 14) | 2nd camp (n = 20) | 3rd camp (n = 17) | 4th camp (n = 14) | 1st vs 2nd | 1st vs 3rd | 1st vs 4th | 2nd vs 3rd | 2nd vs 4th | 3 vs 4th |
| Mean sRPE (a.u.) | 335.4 ± 44.9 | 826.9 ± 125.5 | 894.6 ± 247.6 | 596.9 ± 172.5 | < 0.01** (-4.8 §) | < 0.01** (-2.9 §) | 0.01* (-2.0 §) | 1 (-0.3 #) | 0.01* (1.5 ‡) | < 0.01* (1.3 ‡) |
| Sum sRPE (a.u.) | 1676.8 ± 224.4 | 3266.5 ± 536 | 3635.1 ± 791.8 | 4509.8 ± 1136 | < 0.01** (-3.6 §) | < 0.01** (-3.1 §) | < 0.01** (-3.4 §) | 0.80 (-0.5 #) | < 0.01* (-1.5 ‡) | 0.01* (-0.9 †) |
| Monotony (a.u.) | 7.4 ± 3.1 | 2.7 ± 0.4 | 3.9 ± 3.9 | 3.1 ± 0.8 | < 0.01** (2.3 §) | 0.02* (1.0 †) | < 0.01** (1.8 ‡) | 0.90 (-0.4 #) | 1 (-0.7 †) | 1 (0.3 #) |
| Strain (a.u.) | 12785.2 ± 6313.9 | 8932.5 ± 2096.2 | 14731.9 ± 15508.1 | 13971.2 ± 4994.3 | 1 (0.9 †) | 1 (-0.2 *) | 1 (-0.2 *) | 0.31 (-0.5 #) | 1 (-1.4 ‡) | 1 (0.1 *) |
| Wellness (a.u.) | 20.4 ± 2.5 | 16.9 ± 1.4 | 15.6 ± 3.1 | 16.3 ± 2.5 | 0.01* (1.8 ‡) | < 0.01** (1.6 ‡) | < 0.01** (1.6 ‡) | 0.56 (0.5 #) | 1 (0.3 #) | 1 (-0.2 *) |
| Fatigue (a.u.) | 3.7 ± 0.7 | 2.8 ± 0.3 | 2.8 ± 0.6 | 2.9 ± 0.6 | < 0.01* (1.7 ‡) | < 0.01** (1.4 ‡) | 0.02* (1.2 †) | 1 (0 *) | 1 (-0.2 *) | 1 (-0.2 *) |
| Sleep (a.u.) | 4.4 ± 0.4 | 3.9 ± 0.4 | 3.4 ± 0.7 | 3.7 ± 0.5 | 0.06* (1.2 †) | < 0.01** (1.7 ‡) | 0.02* (1.5 ‡) | 0.02* (0.9 †) | 1 (0.4 #) | 0.29 (-0.5 #) |
| DOMS (a.u.) | 3.9 ± 0.6 | 3.0 ± 0.4 | 2.8 ± 0.8 | 2.8 ± 0.7 | 0.01* (1.8 ‡) | < 0.01** (1.5 ‡) | < 0.01** (1.6 ‡) | 1 (0.3 #) | 1 (0.4 #) | 1 (-0.5 #) |
| Stress (a.u.) | 4.0 ± 0.7 | 3.2 ± 0.4 | 3.1 ± 0.7 | 3.2 ± 0.6 | 0.01* (1.4 ‡) | < 0.01** (1.3 ‡) | 0.02* (1.2 †) | 1 (0.2 *) | 1 (0 *) | 1 (-0.1 *) |
| Mood (a.u.) | 4.4 ± 0.5 | 4.0 ± 0.3 | 3.5 ± 0.8 | 3.6 ± 0.6 | 0.11 (1.0 †) | < 0.01** (1.3 ‡) | < 0.01** (1.4 ‡) | 0.44 (0.8 †) | 0.38 (0.9 †) | 1 (-0.1 *) |

Data are presented as mean and standard deviation or P value and effect size. DOMS = delayed onset muscle soreness.

$p < 0.05$ = *;

$p < 0.01$ = **.

The level of effect size was symbolled trivial (0–0.2) as *,

small (0.2–0.6) as #,

moderate (0.6–1.2) as †,

large (1.2–2.0) as ‡,

very large (> 2.0) as §.

There was a significant difference in mean sRPE among the OTCs [$F(3, 64) = 9.9$, $p < 0.01$] (Table 3). The coefficient of variation (CV) of mean sRPE was lowest in the 1st OTC (17.4 ± 10.7%) and largest in the 3rd OTC (39.8 ± 19.8%). The qualitative magnitude was very large when the 1st OTC was compared with the 2nd OTC, 3rd, and 4th OTC, respectively.

## General wellness score

One-way ANOVA revealed a significant difference in wellness [$F(3, 64) = 11.6$, $p < 0.01$], fatigue [$F(3, 64) = 8.6$, $p < 0.01$], sleep [$F(3, 64) = 9.6$, $p < 0.01$], DOMS [$F(3, 64) = 9.6$, $p < 0.01$], stress [$F(3, 64) = 5.7$, $p < 0.01$], and mood [$F(3, 64) = 9.2$, $p < 0.01$]. Lowest wellness scores were found in the 3rd OTC (wellness = 15.6 ± 3.1 a.u.; fatigue = 12.8 ± 0.6 a.u.; sleep = 3.4 ± 0.7 a.u.; DOMS = 2.8 ± 0.8 a.u.; stress = 3.1 ± 0.7 a.u.; mood = 3.5 ± 0.8 a.u.) whereas largest wellness elements were found in the 1st OTC (wellness = 20.4 ± 2.5 a.u.; fatigue = 3.7 ± 0.7 a.u.; sleep = 4.4 ± 0.4 a.u.; DOMS = 3.9 ± 0.6 a.u.; stress = 4.0 ± 0.7 a.u.; mood = 4.4 ± 0.5 a.u.). Measures of magnitude demonstrated large ES in all wellness elements when the 1st OTC was compared with others OTC's, except for fatigue and stress in the 1st OTC vs 4th OTC pairwise comparison and mood in the 1st OTC vs 2nd OTC pairwise comparison (Table 2).

There was also a significant difference in sleep [$F(3, 64) = 3.0$, $p = 0.36$] and mood [$F(3, 64) = 5.2$, $p = 0.03$]. The CV of wellness elements varied from camp to camp (wellness = 8.7 ± 4.9%

**Table 3. Coefficient of variation of internal load (session rating of perceived exertion) and wellness status (wellness, fatigue, delayed onset muscle soreness, sleep, stress, and mood) in four overseas training camps.**

| | Overseas training camps | | | | P value (Effect size) | | | | | |
|---|---|---|---|---|---|---|---|---|---|---|
| | 1st camp (n = 14) | 2nd camp (n = 20) | 3rd camp (n = 17) | 4th camp (n = 14) | 1st vs 2nd | 1st vs 3rd | 1st vs 4th | 2nd vs 3rd | 2nd vs 4th | 3 vs 4th |
| Mean sRPE (%) | 17.4 ± 10.7 | 37.5 ± 5.5 | 39.8 ± 19.8 | 34.2 ± 9.4 | < 0.01** (-2.4 §) | < 0.01** (-1.3 ‡) | < 0.01** (-1.6 ‡) | 1 (-0.2 *) | 1 (0.4 #) | 1 (0.3 #) |
| Wellness (%) | 11.1 ± 6.5 | 9 ± 3.9 | 8.7 ± 4.9 | 11.8 ± 8.8 | 1 (0.4 #) | 1 (0.4 #) | 1 (-0.1 *) | 1 (-0.4 #) | 1 (-0.4 #) | 0.98 (-0.4 #) |
| Fatigue (%) | 22.2 ± 14 | 22 ± 7.9 | 16.6 ± 13.2 | 18.7 ± 12.4 | 1 (0.0 *) | 1 (0.4 #) | 1 (0.3 #) | 1 (0.5 #) | 1 (0.3 #) | 1 (-0.2 *) |
| Sleep (%) | 8 ± 7.1 | 10.6 ± 10.4 | 18.1 ± 11 | 14.9 ± 11.7 | 1 (-0.3 #) | 0.05 (-1.0 †) | 0.47 (-0.7 †) | 0.19 (-0.7 †) | 1 (-0.4 #) | 1 (0.3 #) |
| DOMS (%) | 16.5 ± 8 | 16.1 ± 12.1 | 19.3 ± 10.3 | 21.9 ± 11.4 | 1 (0.0 *) | 1 (-0.3 #) | 1 (-0.5 #) | 1 (-0.3 #) | 0.78 (-0.5 #) | 1 (-0.2 *) |
| Stress (%) | 15.3 ± 10.5 | 9 ± 11.7 | 19.3 ± 14.1 | 12.6 ± 14.6 | 0.96 (0.6 #) | 1 (-0.3 #) | 1 (0.2 *) | 0.11 (-0.8 †) | 1 (-0.3 #) | 0.9 (0.5 #) |
| Mood (%) | 6.8 ± 5.3 | 6 ± 7.8 | 20.6 ± 19.1 | 12.6 ± 11.7 | 1 (0.1 *) | 0.02* (-0.9 †) | 1 (-0.6 #) | < 0.01** (-1.0 †) | 0.74 (-0.7 †) | 0.47 (0.5 #) |

Data are presented as mean and standard deviation or p value and effect size. DOMS = delayed onset muscle soreness.

p < 0.05 = *;

p < 0.01 = **.

The level of effect size was symbolled trivial (0–0.2) as *,

small (0.2–0.6) as #,

moderate (0.6–1.2) as = †,

large (1.2–2.0) as ‡,

very large (> 2.0) as §.

~ 11.8 ± 8.8%; fatigue = 16.6 ± 13.2% ~ 22.2 ± 14%; sleep = 8 ± 7.1 ~ 18.1 ± 11%; DOMS = 16.1 ± 12.1 ~21.9 ± 11.4%; stress = 9 ± 11.7 ~ 19.3 ± 14.1%; mood = 6 ± 7.8 ~ 20.6 ± 19.1%). The magnitude of ES varied from trivial to moderate among the comparisons (from 0 ~ -1.0) (Table 3).

### Linear regression analysis

The mean and sum sRPE negatively correlated with all wellness elements ($p < 0.001$). The mean sRPE shows a range of $r$ value from -0.575 (wellness score) to -0.439 (mood). Furthermore, the mean sRPE shows a ragne of $r$ value from -0.559 (wellness score) to -0.410 (stress). However, training monotony ($r = 0.034–0.216$) and training strain ($r = -0.097–0.302$) showed no relationship with all wellness elements, excepted training monotony vs fatigue ($p = 0.042$), training monotony vs fatigue ($p = 0.046$), training strain vs sleep ($p = 0.007$) (see Fig 1).

### Discussion

The primary purpose of this study was to identify characteristics of the RPE-based measures and wellness status in elite futsal players during task-specific OTCs. The secondary purpose was to investigate the relationship between RPE-based measures and wellness status during futsal OTCs. Based on the observations in mean sRPE and sum sRPE, training monotony, training strain, and wellness scores during OTCs are task-dependent in elite futsal players. Additionally, the CV of sRPE is lower during game-based OTCs but larger during training-based and pre-tournament OTCs. The magnitude of CV of wellness variables ranged from trivial to moderate, indicating no large daily fluctuation in wellness status during futsal OTCs. These findings supported our first hypothesis. The secondary finding in the present study

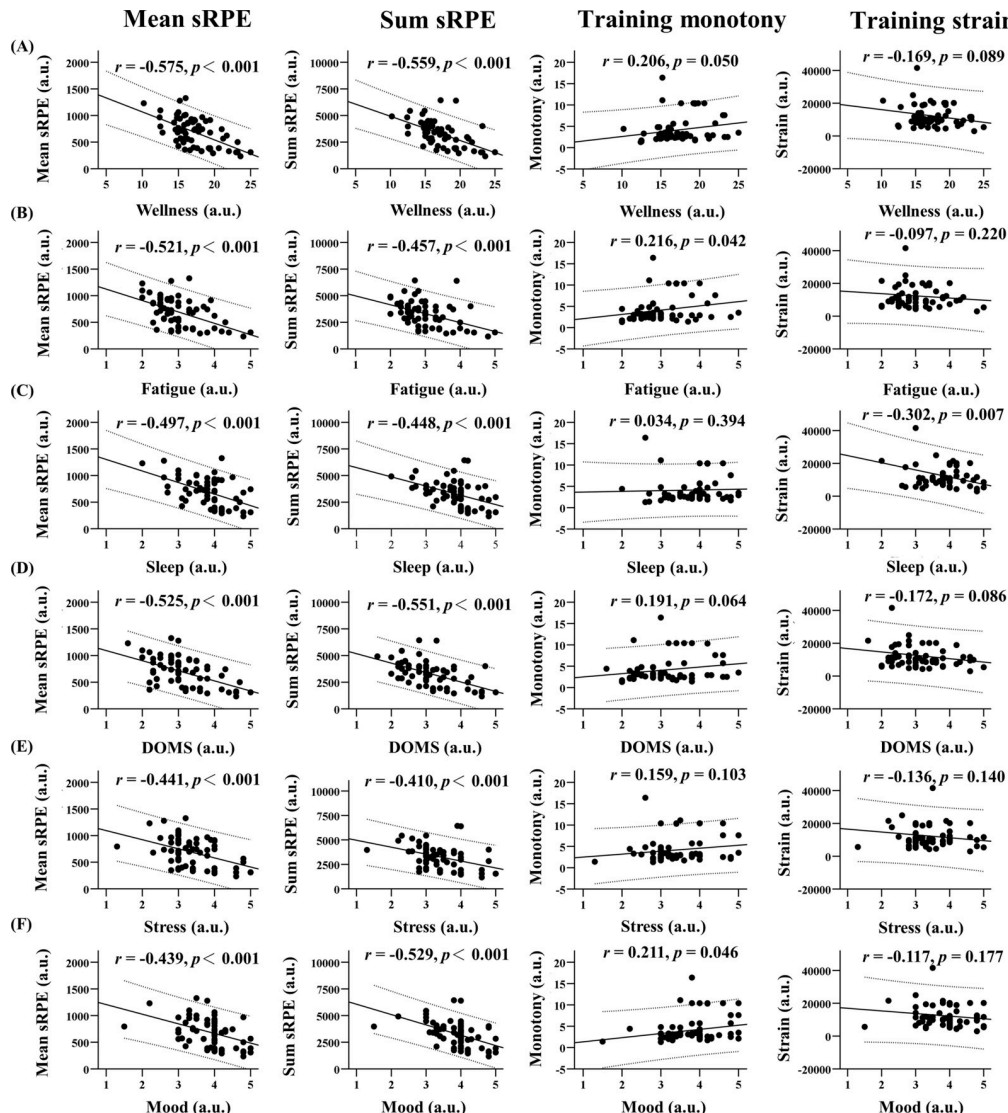

**Fig 1. The linear regression analysis between perceived measures of internal loads and wellness status during futsal overseas training camps.** Columns illustrate relationships between mean session rating of perceived exertion/ sum session rating of perceived exertion/ training monotony/training strain (from most left column to most right column in the continuous sequence) and all wellness components: A) relationship with wellness scores, B) relationship with fatigue, C) relationship with sleep, D) relationship with delayed onset muscle soreness, E) relationship with stress, F) relationship with mood.

showed a significant negative relationship between mean and sum sRPE scores and all wellness components. Conversely, training monotony and training strain had no relationship with all wellness components. The later finding rejected our secondary hypothesis.

The primary finding in the present study showed a large variation of sRPE and training monotony across the OTCs. This observation indicates a characteristic of task-dependent TL during each OTC. The sRPE in the 1st OTC was significantly lower than that of 2nd, 3rd, and 4th OTC (1st OTC = 1680.47 ± 216.70 a.u. vs 2nd OTC = 3266.45 ± 536.01 a.u., 3rd OTC = 3635.06 ± 791.85 a.u., 4th OTC = 4509.29 ± 1135.97 a.u.). In this study, the sum sRPE scored in each OTC was related to the intensity of sessions/match and duration. The 1st OTC consisted mainly of game-based tasks to test the competitive level of the selected players.

While, the 2nd and 3rd OTC focused on testing playing systems, fitness training, the strength of the bench depth for new players, and the implementation of team strategies. The 4th OTC was the final stage of pre-tournament preparation to simulate the competition environment and test game strategies before the final competition. Studies utilizing sRPE to quantify TL have reported ranges between 4–7 points for weekly mean RPE during a futsal preseason period [31]. However, the players' RPE scores varied from camp to camp in our study (i.e. 1st OTC RPE = 2–5 points; 2nd OTC RPE = 3–7 points, 3rd OTC RPE = 2–9 points, 4th OTC RPE = 3–10 points). The discrepancies of RPE scores reported between the futsal preseason (consistency/progression of RPE) and the short-term futsal OTCs (variability of RPE) indicate the task-dependent characteristics of TL exhibited during each OTC.

Moreover, very large ES of mean and sum sRPE and training monotony were found when the 1st OTC was compared to the other OTCs. This discrepancy can be attributed to four consecutive matches held within the 1st OTC which were used to evaluate tactical plans for future matches. Therefore, only one training session was conducted in this OTC. Training monotony and training strain are variants of sRPE and help quantify weekly TL variability which is associated with the risk of injury and overtraining [29]. In professional soccer players, high intensity TL during the preseason period contributes to greater training monotony and training strain and is associated with the perceived measure of muscle fatigue and pain during the competitive season [32]. Furthermore, previous longitudinal studies assessed by accelerometry-based TL showed that decreasing training monotony is linked to an increase in training strain over a playing season in professional soccer players [24]. Training monotony and training strain (measured by GPS-derived accelerometer) tended to increase at the beginning and late stages of the season while decreasing during the middle portion of the season in professional soccer players [28]. In Futsal, Stochi de Oliveira and Borin [33] observed that Brazilian elite players demonstrated low training monotony (1.4–1.7 A.U.) and training strain (>5000 A.U.) during an entire futsal season, indicating a high degree of TL variability. In the present study, the extremely large training monotony and training strain during the 1st OTC (6 days) is related to the congested match schedule. In contrast, the 4th OTC (10 days) had only one scheduled match, followed by a rest/training day. The discrepancy of workload patterns between the professional season and national team TC highlights the influence of task-dependent profiles on TL. However, coaches should carefully consider such evidence since higher training monotony, and training strain levels may increase injury risk [29].

In this study, TL variability was detected in the CV estimations. The CV of sRPE was lower during the 1st OTC but was higher during the 2nd, 3rd, 4th OTCs. This phenomenon may be related to the same level of opponents played across matches in the 1st OTC. Intra-day variation of sRPE is a training marker to help understand fluctuations in individual TL responses. This finding is supported by the large training monotony found in the 1st OTC. As observed in the large SD in the 3rd OTC, a large intra-individual variation in TL may occur during intensive training-based camps. This intra-player variation should be considered in the weekly periodization of training stimuli since training intensity variability is one strategy to avoid monotony. Additionally, the impact of training intensity between players should be considered in future research to evaluate the acute responses to different levels of training stimuli.

As demonstrated in Tables 1 and 2, a larger ES of wellness scores was identified when the mean value was used to determine qualitative magnitude among the OTCs. On the other hand, trivial or small ES of wellness scores were found when CV was used for comparisons. These findings indicate that the daily fluctuation in wellness status was minor during futsal OTCs. Coincidentally, our previous observation found no change in daily wellness status and resting HR in senior national futsal players during a 5-day OTC with two friendly matches [11]. This suggests that although there exists a relationship between TL and wellness, other

contextual factors and intra-personal behaviors may modulate wellness scores. In fact, a recent systematic review revealed a weak relationship between TL and wellness. This suggests that scores of wellness are influenced by many factors outside of just the TL imposed [22].

The linear regression analysis demonstrated a negative relationship between mean and sum sRPE and all wellness variables but not between training monotony and training strain. Our recent study supports these findings, which demonstrated a negative relationship between TL and wellness scores during invitation tournaments and training camps with high TL [8]. It is well known that TL and wellness status are primary factors that affect the psychophysiological health of athletes and the effectiveness of training adaptations [34]. Both measures can be used as a simple tool to examine the wellness conditions during TC [32, 35]. For example, higher TL and lower wellness scores are associated with lower cardiac-vagal tone and vice versa [36, 37].

There are three main limitations of the current study. Firstly, the traveling schedule was tabulated based on the convenience of flights. The discrepancy of traveling time among the OTC may have contributed to potential impacts on players' psychometric status and circadian rhythm on the first day. Secondly, the training days varied from camp to camp. Although an elite coach took charge of the four OTCs, individual adaptation may vary among players [31]. Thirdly, the players experienced different playing times during friendly matches. This playing time variation may influence the reported TL scores on the friendly match day and vice versa.

Regarding practical implications, the negative association between sRPE and wellness status during short-term OTCs reflects the need to incorporate a measure of psychological exertion to wellness ratio in future investigations and training environments. Such implantations can help examine and track physical strain and wellbeing status during club/school days or domestic training camps prior to an OTC. Coaches and strength and conditioning practitioners are encouraged to utilize a comprehensive evaluation methods to monitor the daily fluctuation of psychophysiological responses during futsal OTCs.

For the future studies, it is recommended to futher explore the relationship between perceived measure of TL and wellness status during the match day and the day after the match during OTCs. Such information can advance our current understanding of managing individual variations of health status and players exposture to physical exertion during the microcycle of camps.

## Conclusions

In conclusion, the perceived responses in training exertion and wellness scores are task-dependent during OTCs in elite futsal players. Utilizing mean/sum sRPE and wellness assessments to monitor psychophysiological health during short-term OTCs is recommended. The RPE-based training monotony and strain scores are independent markers of fatigue, DOMS, sleep, stress, and mood during futsal OTCs. Consequently, an integrated approach to using perceived measures of TL and assessments of wellness status provides efficient information in relation to training stress and wellness status in elite futsal players during OTCs.

## Supporting information

**S1 Data.**
(SAV)

**S2 Data.**
(SAV)

## Acknowledgments

The authors would like to thank the head coach Adil Amarante, team members, and U-20 Chinese Taipei futsal team players who volunteered for this study.

## Author Contributions

**Conceptualization:** Yung-Sheng Chen, Filipe Manuel Clemente, Cheng-Deng Kuo.

**Data curation:** Jeffrey Cayaban Pagaduan, Yu-Xian Lu, Chia-Hua Chien, Yi-Wen Chiu.

**Formal analysis:** Filipe Manuel Clemente.

**Funding acquisition:** Cheng-Deng Kuo.

**Investigation:** Chia-Hua Chien, Yi-Wen Chiu.

**Methodology:** Yung-Sheng Chen, Filipe Manuel Clemente, Jeffrey Cayaban Pagaduan, Zachary J. Crowley-McHattan, Yu-Xian Lu, Chia-Hua Chien, Pedro Bezerra.

**Project administration:** Yung-Sheng Chen, Yu-Xian Lu.

**Resources:** Pedro Bezerra.

**Software:** Jeffrey Cayaban Pagaduan, Yu-Xian Lu, Chia-Hua Chien, Yi-Wen Chiu.

**Supervision:** Yung-Sheng Chen, Yi-Wen Chiu, Cheng-Deng Kuo.

**Visualization:** Pedro Bezerra.

**Writing – original draft:** Yung-Sheng Chen, Filipe Manuel Clemente, Zachary J. Crowley-McHattan, Pedro Bezerra, Cheng-Deng Kuo.

**Writing – review & editing:** Yung-Sheng Chen, Filipe Manuel Clemente, Zachary J. Crowley-McHattan, Pedro Bezerra, Cheng-Deng Kuo.

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
