## [Decision Letter · Decision Letter 0]

31 Jan 2022

PONE-D-21-37878Relationships between perceived internal training loads and psychometric wellness during overseas futsal training campsPLOS ONE

Dear Dr. Kuo,

Thank you for submitting your manuscript to PLOS ONE. After careful consideration, we feel that it has merit but does not fully meet PLOS ONE’s publication criteria as it currently stands. Therefore, we invite you to submit a revised version of the manuscript that addresses the points raised during the review process.

Please, address point-to-point all reviewers' issues.

We look forward to receiving your revised manuscript.

Kind regards,

Luca Paolo Ardigò, Ph.D.

Academic Editor

PLOS ONE

Journal Requirements:

 (This work was supported by a grant VGHUST96-P1-06 from Taipei Veterans General Hospital, and a grant MOST-103-2410-H-075-001 from the Ministry of Science and Technology, Taiwan.)

(The authors would like to thank the head coach Adil Amarante, team members, and U-20 Chinese Taipei futsal team players who volunteered for this study.

This work was supported by a grant VGHUST96-P1-06 from Taipei Veterans General Hospital, and a grant MOST-103-2410-H-075-001 from the Ministry of Science and Technology, Taiwan.)

(This work was supported by a grant VGHUST96-P1-06 from Taipei Veterans General Hospital, and a grant MOST-103-2410-H-075-001 from the Ministry of Science and Technology, Taiwan.)

Additional Editor Comments:

Please, address point-to-point all reviewers' issues.

Reviewers' comments:

Reviewer's Responses to Questions

**Comments to the Author**

1. Is the manuscript technically sound, and do the data support the conclusions?

Reviewer #1: Yes

Reviewer #2: Yes

2. Has the statistical analysis been performed appropriately and rigorously? 

Reviewer #1: Yes

Reviewer #2: Yes

3. Have the authors made all data underlying the findings in their manuscript fully available?

Reviewer #1: No

Reviewer #2: Yes

4. Is the manuscript presented in an intelligible fashion and written in standard English?

Reviewer #1: Yes

Reviewer #2: Yes

5. Review Comments to the Author

Reviewer #1: PONE-D-21-37878

ABSTRACT

Line 47: maybe consider rethinking the use of “training load”. I recommend reading the article “Misuse of the term ‘load’ in sport and exercise science” - https://www.sciencedirect.com/science/article/pii/S1440244021002127

Line 50: recovery status – how the process will be monitored? Maybe more detail here.

Line 53: add the eligibility criteria for the included players

Line 55: maybe add the details of the wellness questionnaire or the items assessed.

Lines 57-61: add statistical values to support the statements

INTRODUCTION

Line 93: there is a perceived measure of internal training load? Or perceived effort/exertion? CR-10 Borg measure the perceived intensity and not load. https://www.sciencedirect.com/science/article/pii/S1440244021002127

Lines 93-94: support the statement about the cognitive awareness

Lines 94-96: Maybe some description about validity and reliability of the scale can be good.

Line 98: add examples of wellness questionnaires, validity and reliability. Which questionnaire?

Lines 99-100: maybe consider this systematic review - Duignan, C., Doherty, C., Caulfield, B., & Blake, C. (2020). Single-Item Self-Report Measures of Team-Sport Athlete Wellbeing and Their Relationship With Training Load: A Systematic Review. Journal of Athletic Training, 55(9), 944-953.

METHODS

Line 124: contextualize the periods of data collection (i.e., early season, mid season, end season) and the exact dates

Line 135: add the a priori sample size using G*Power or a similar software

Line 144: how many times the same player participated in the OTC? Maybe a table with the N of sessions per each players would be important

Line 165: add the exact question made to the players prior to score is provided (example: how intense was the session?)

Line 178: same as above (i.e., exact question) for the case of wellness.

RESULTS

Tables: add p-value as well and not the magnitude-based inference.

The figure has a low-resolution. Must be improved the quality.

DISCUSSION

Generally well-written. I would like to suggest more discussion about further research and more practical recommendations for using not only in OTC contexts.

Reviewer #2: General comments:

The study investigated the effects of different OTC on TL and well being measures as well as their relationships. The study is of interest and the authors did good job. There are some issues to be addressed, and a general English revision is required before the formal acceptance of the paper.

Specific comments

Abstract

L49: the authors said that the relationship between TL and recovery status has not been established during short-term overseas training camps (OTC). However, previous study (Lu et al., 2018) has investigated such relationship during OTC? Any clarifications? Please remove it or just say that it was not extensively investigated to not neglect previous research.

- Please add effect size’s values for different comparisons and r values for correlations.

Introduction

This part is well written by the authors.

Why the authors did not report that one previous study also investigated the relationship between TL and recovery state in OTC (https://www.termedia.pl/Training-load-and-recovery-status-during-a-short-term-overseas-training-camp-in-Taiwan-futsal-players,132,35932,0,0.html). I think it is important for your paper to clear for readers what is different from this one and this strengthens your rationale.

Materials and Methods

Participants

L138: “called up” may be appropriate to change to “recruited “

- A sample size calculation is required here, please add it.

- Please provide psychometric properties for measures used

Discussion

The discussion is written well.

Figure

I would ask the authors to change the figure as it is not clear.

6. PLOS authors have the option to publish the peer review history of their article (what does this mean?). If published, this will include your full peer review and any attached files.

Reviewer #1: No

Reviewer #2: No

---

## [Author Response · Author response to Decision Letter 0]

16 Feb 2022

Plos One

Manuscript number: PONE-D-21-37878

Title: Relationship between Relationships between perceived internal training loads and psychometric wellness during overseas futsal training camps

Dear Reviewers:

Thank you very much for your suggestions and the valuable comments. We have revised the manuscript according to the reviewers’ comments. We provide point-by-point responses (in RED color) to the reviewers’ comments below. Revision in responses to each question/concern raised by the reviewers are noted with tracked changes in the manuscript.

REVIEWER 1

ABSTRACT

Line 47: maybe consider rethinking the use of “training load”. I recommend reading the article “Misuse of the term ‘load’ in sport and exercise science” - https://www.sciencedirect.com/science/article/pii/S1440244021002127

Response: Thanks for your positive comments and support. We appreciate the need to correct the description to use training load in the revision after reading the reference. We have revised the description in the revision to highlight this issue. Please refer to revision for the changes. Thanks again to update the valuable information with us.

Line 50: recovery status – how the process will be monitored? Maybe more detail here.

Response: Thank you for your comment. We have revised and added further description in the sentence in Line 59-60.

Line 53: add the eligibility criteria for the included players

Response: Thank you for your valuable comment. We added a sentence in line 54-55.

Line 55: maybe add the details of the wellness questionnaire or the items assessed.

Response: Thank you for your suggestion. We have revised the sentence in line 59-61. 

Lines 57-61: add statistical values to support the statements

Response: Thank you for your valuable comment. We have added ES and r value in the abstract as per your comment. 

INTRODUCTION

Line 93: there is a perceived measure of internal training load? Or perceived effort/exertion? CR-10 Borg measure the perceived intensity and not load. https://www.sciencedirect.com/science/article/pii/S1440244021002127

Response: We fully agree with the reviewer’s comment, according to the statements reported by Staunton, et al. (2021) . We decided to revise all text related to the training load information throughout the manuscript to address the reviewer’s concerns.

Lines 93-94: support the statement about the cognitive awareness

Response: Thanks for your comments about this matter. To avoid confusing to reading, we removed the cognitive awareness in the revision.

Lines 94-96: Maybe some description about validity and reliability of the scale can be good.

Response: Thank you for your valuable comment. We have added additional information in the revision, line 107-108. Thanks. 

Line 98: add examples of wellness questionnaires, validity and reliability. Which questionnaire?

Response: Thank you for your valuable comment. We have added a statement regarding validity and reliability to the revised manuscript. Line 112-113.

Lines 99-100: maybe consider this systematic review - Duignan, C., Doherty, C., Caulfield, B., & Blake, C. (2020). Single-Item Self-Report Measures of Team-Sport Athlete Wellbeing and Their Relationship With Training Load: A Systematic Review. Journal of Athletic Training, 55(9), 944-953.

Response: Thank you for your valuable comment. We have added a sentence “The intensity of TL and recovery status depends upon the types and objectives of the sessions.” to support the statement. The reference Duignan et al.,2020 was added as reference numbered 20. Line 114-115.

METHODS

Line 124: contextualize the periods of data collection (i.e., early season, mid season, end season) and the exact dates

Response: Thank for your point. We have added the exact dates of each training camp in the revision. Line 146-148.

Line 135: add the a priori sample size using G*Power or a similar software

Response: Thank you for your valuable comment. We agree with the reviewer that sample size estimation is critical to examine the power of the results. However, this study is an observational study, based on the training schedule and team preparation of a national team. This study was based on a sample of convenience and thus the number of players varied from camp to camp. We will keep this factor in mind for preparing overseas training camps in the future.

Line 144: how many times the same player participated in the OTC? Maybe a table with the N of sessions per each players would be important

Response: Thank you for your comments to improve the quality of the manuscript. The schedule of overseas training camps was tabulated in table 1. Please refer to information in line 160-162 of the revised manuscript. 

Line 165: add the exact question made to the players prior to score is provided (example: how intense was the session?)

Response: Thank you for your valuable comment. We revised the sentence as “After each training session, the team sports trainer asked the players “how hard was your training session?” before the players reported an individual RPE score. Line 187-189.

Line 178: same as above (i.e., exact question) for the case of wellness.

Response: Thank you for your valuable comment. We added two sentences “The team sports trainer asked the players “how do you feel about the level of fatigue status, sleep quality, muscle soreness, mental stress, and mood?” Afterward, the players reported the scores of each item individually.”. Line 199-201.

RESULTS

Tables: add p-value as well and not the magnitude-based inference.

Response: Thank you for your this suggestion. We performed one-way ANOVA to compare the post-hoc analysis. The F ratio and p values have been added in the revisions. We also retained the effect size in the table due to unequal number of participants among the camps.

The figure has a low-resolution. Must be improved the quality.

Response: Thank you for your comment. We apologized for the low-resolution of figure 1 in the manuscript. After double checking, the figure 1 upload to the submission matches the resolution requirement of Plos One (300 dpi). However, our originally submission was a JPG file. We have converted to a TIFF file as the journal requires. 

DISCUSSION

Generally well-written. I would like to suggest more discussion about further research and more practical recommendations for using not only in OTC contexts.

Response: Thank you for your positive comments and support. We have revised some sentences to improve the quality of discussion within the manuscript according to the reviewers point. Please see our revision. 

REVIEWER 2

The study investigated the effects of different OTC on TL and well being measures as well as their relationships. The study is of interest and the authors did good job. There are some issues to be addressed, and a general English revision is required before the formal acceptance of the paper.

Response: Thank you very much for your supportive comment and valuable suggestions. We have improved the quality of the paper, according to the reviewers’ comments. Additionally, a native English speaker with good knowledge in sports sciences has provided feedback and proofread the revised manuscript. 

Specific comments

Abstract

L49: the authors said that the relationship between TL and recovery status has not been established during short-term overseas training camps (OTC). However, previous study (Lu et al., 2018) has investigated such relationship during OTC? Any clarifications? Please remove it or just say that it was not extensively investigated to not neglect previous research.

Response: Thank you for the suggestion. We have now revised the sentence, according to your comments in lines 49-51. 

- Please add effect size’s values for different comparisons and r values for correlations.

Response: Thank you for your valuable comment. We added ES and r value of correlation output in the abstract as per your comment. 

Introduction

This part is well written by the authors.

Response: Thank for your positive comment.

Why the authors did not report that one previous study also investigated the relationship between TL and recovery state in OTC (https://www.termedia.pl/Training-load-and-recovery-status-during-a-short-term-overseas-training-camp-in-Taiwan-futsal-players,132,35932,0,0.html). I think it is important for your paper to clear for readers what is different from this one and this strengthens your rationale.

Response: Thank for your valuable comment. We added a sentence to address your concern. Line 125-127.

Materials and Methods

Participants

L138: “called up” may be appropriate to change to “recruited “

Response: Revised accordingly. Line 159.

- A sample size calculation is required here, please add it.

Response: Thank for your valuable comment. We agree with the reviewer that sample size estimation is critical to examine the power of the results. However, this study is an observational study, based on the training schedule and team preparation of a national team. This study was therefore based on a sample of convenience and thus the number of players varied from camp to camp. We will keep this factor in mind for preparing overseas training camps in the future.

- Please provide psychometric properties for measures used

Response: We apologize for the misleading statement for psychometric properties. To avoid the potential for confusion, we have remove the term “psychometric” and in its place inserted wellness status in the revision. Line 195-203.

Discussion

The discussion is written well.

Response: Thank you for your positive comment.

Figure

I would ask the authors to change the figure as it is not clear.

Response: Thank you for your comment. We apologized for the low-resolution of figure 1 in the manuscript. After double checking, the figure 1 upload to the submission matches the resolution requirement of Plos One (300 dpi). However, our originally submission was a JPG file. We have converted to a TIFF file as the journal requires.

---

## [Decision Letter · Decision Letter 1]

18 Mar 2022

PONE-D-21-37878R1Relationships between perceived measures of internal load and wellness status during overseas futsal training campsPLOS ONE

Dear Dr. Kuo,

Thank you for submitting your manuscript to PLOS ONE. After careful consideration, we feel that it has merit but does not fully meet PLOS ONE’s publication criteria as it currently stands. Therefore, we invite you to submit a revised version of the manuscript that addresses the points raised during the review process.

Please, one further effort to address Reviewer 1's minor issues.

We look forward to receiving your revised manuscript.

Kind regards,

Luca Paolo Ardigò, Ph.D.

Academic Editor

PLOS ONE

Journal Requirements:

Additional Editor Comments (if provided):

Please, one further effort to address Reviewer 1's minor issues.

Reviewers' comments:

Reviewer's Responses to Questions

**Comments to the Author**

1. If the authors have adequately addressed your comments raised in a previous round of review and you feel that this manuscript is now acceptable for publication, you may indicate that here to bypass the “Comments to the Author” section, enter your conflict of interest statement in the “Confidential to Editor” section, and submit your "Accept" recommendation.

Reviewer #1: (No Response)

Reviewer #2: All comments have been addressed

2. Is the manuscript technically sound, and do the data support the conclusions?

Reviewer #1: Yes

Reviewer #2: Yes

3. Has the statistical analysis been performed appropriately and rigorously? 

Reviewer #1: Yes

Reviewer #2: Yes

4. Have the authors made all data underlying the findings in their manuscript fully available?

Reviewer #1: No

Reviewer #2: Yes

5. Is the manuscript presented in an intelligible fashion and written in standard English?

Reviewer #1: Yes

Reviewer #2: Yes

6. Review Comments to the Author

Reviewer #1: ABSTRACT

Line 49: detail the meaning of “recovery status”

Lines 51-52: would be better to add a description of the outcomes

Line 53: add the sample of training sessions and matches analyzed.

Lines 55-56: add the outcomes coming from the questionnaire.

Lines 57-61: add statistical values to support the statements.

INTRODUCTION

Lines 73-74: maybe add some details regarding the typical locomotor demands and intermittency of them.

Lines 94-96: Borg’s scale does not allow measuring training load. In fact, is a measure of training intensity. Training load is a combination of intensity and volume.

Lines 93-98: it would be important to strengthen the rationale of this paragraph with the Gabbett, T. J., Nassis, G. P., Oetter, E., Pretorius, J., Johnston, N., Medina, D., ... & Ryan, A. (2017). The athlete monitoring cycle: a practical guide to interpreting and applying training monitoring data. British Journal of Sports Medicine, 51(20), 1451-1452.

The terms wellness and recovery status are not the same. Recovery status involves more than well-being measures, namely the functional status. I would like to suggest avoiding the term recovery status.

The introduction section synthetizes the current state of the art. Nerveless, some references can help to better contextualizing these sections (e.g., Almeida, et. al. 2019.Coach decision-making in Futsal: from preparation to competition. International Journal of Performance Analysis in Sport.; Sarmento, H., Bradley, P., Travassos, B. (2015). The Transition from Match Analysis to Intervention: Optimising the Coaching Process in Elite Futsal. International Journal of Performance Analysis in Sport)

METHODS

Line 124: a timeline would be important adding information about the occurrence of the events.

Table 1: would be important to add information about how many training camps occurred for each type.

Line 122: sRPE was not introduced before

Lines 171 and 178: add the final main outcomes used for further data treatment. Moreover, add information about how statistical procedures were conducted (e.g., using average of training camps? Average of each day of training camps?)

Line 182: add the test to measure the homogeneity. Both were observed?

RESULTS

Figure 1. resolution of the figure is low.

Line 228: add in-text description of the information about statistical values associated with the linear regression.

DISCUSSION

Generally well-written. However, future research should be described after the study limitations.

Reviewer #2: I congratulate the authors as they addressed almost of my comments satisfactorly. I recommend the paper to be accepted.

7. PLOS authors have the option to publish the peer review history of their article (what does this mean?). If published, this will include your full peer review and any attached files.

Reviewer #1: No

Reviewer #2: No

---

## [Author Response · Author response to Decision Letter 1]

30 Mar 2022

Manuscript number: PONE-D-21-37878

Title: Relationships between perceived measures of internal load and wellness status during overseas futsal training camps

Dear Reviewers:

Thank you very much for your suggestions and the valuable comments in the second round of review. We have revised/improved the manuscript according to your comments. We provide point-by-point responses (in RED color) to the reviewers’ comments below. Revision in responses to each question/concern raised by the reviewers are noted with tracked changes in the manuscript.

REVIEWER 1

ABSTRACT

Line 49: detail the meaning of “recovery status”

Responses: The term “recovery status” was changed to “wellness status” throughout the manuscript.

Lines 51-52: would be better to add a description of the outcomes

Responses: Done.

Line 53: add the sample of training sessions and matches analyzed.

Responses: Done.

Lines 55-56: add the outcomes coming from the questionnaire.

Responses: We have revised the abstract entirely. We believe the current form is acceptable, according to your comment.

Lines 57-61: add statistical values to support the statements.

Responses: We have revised the abstract entirely. We believe the current form is acceptable, according to your comment.

INTRODUCTION

Lines 73-74: maybe add some details regarding the typical locomotor demands and intermittency of them.

Responses: This is a great point to enrich the statement. We added few sentences to this address this point. Please refer to line 90-92.

Lines 94-96: Borg’s scale does not allow measuring training load. In fact, is a measure of training intensity. Training load is a combination of intensity and volume.

Responses: Dear reviewer. We have corrected the statements in the first revision. We believe the paragraph between line 112-118 is fully compliant with your point here.

Lines 93-98: it would be important to strengthen the rationale of this paragraph with the Gabbett, T. J., Nassis, G. P., Oetter, E., Pretorius, J., Johnston, N., Medina, D., ... & Ryan, A. (2017). The athlete monitoring cycle: a practical guide to interpreting and applying training monitoring data. British Journal of Sports Medicine, 51(20), 1451-1452.

Responses: Added a sentence to strengthen the rationale of facilitating overseas training camps. Please refer to line 108-111.

The terms wellness and recovery status are not the same. Recovery status involves more than well-being measures, namely the functional status. I would like to suggest avoiding the term recovery status.

Responses: Thanks for your valuable comment to this point. We entirely changed the “recovery status” to “wellness status”. 

The introduction section synthetizes the current state of the art. Nerveless, some references can help to better contextualizing these sections (e.g., Almeida, et. al. 2019.Coach decision-making in Futsal: from preparation to competition. International Journal of Performance Analysis in Sport.; Sarmento, H., Bradley, P., Travassos, B. (2015). The Transition from Match Analysis to Intervention: Optimising the Coaching Process in Elite Futsal. International Journal of Performance Analysis in Sport)

Responses: Thank for the suggestion to enrich the introduction. We added information in the last paragraph of the introduction, between line 141-144.

METHODS

Line 124: a timeline would be important adding information about the occurrence of the events.

Responses: The data of training camps has been descripted between line 158-.161. We added information for a time schedule in Table 1 for clarity.

Table 1: would be important to add information about how many training camps occurred for each type.

Responses: Done.

Line 122: sRPE was not introduced before

Responses: We moved the subsection Data Collection to line 220.

Lines 171 and 178: add the final main outcomes used for further data treatment. Moreover, add information about how statistical procedures were conducted (e.g., using average of training camps? Average of each day of training camps?)

Responses: We added a sentence “The average of individual values in each training camp was used for subsequent data analysis.” between line 196-197 for clarity.

Line 182: add the test to measure the homogeneity. Both were observed?

Responses: We did not included homogeneity test for one way ANOVA test. However, we used a post hoc test with Bonferroni adjustment for the pairwise comparison. We added this information in line 231 for clarity.

RESULTS

Figure 1. resolution of the figure is low.

Responses: We increase the resolution to 600 dpi. 

Line 228: add in-text description of the information about statistical values associated with the linear regression.

Responses: Added accordingly. See line 280-285.

DISCUSSION

Generally well-written. However, future research should be described after the study limitations.

Responses: Done. Please refer to line 384-388 in the revision.

Dear Reviewer:

Thank you very much for your suggestions and the valuable comments. 

REVIEWER 2

I congratulate the authors as they addressed almost of my comments satisfactorly. I recommend the paper to be accepted.

Responses: Thank you very much for your time and effort to improve the quality of the manuscript. Thanks again for your support.

---

## [Decision Letter · Decision Letter 2]

5 Apr 2022

Relationships between perceived measures of internal load and wellness status during overseas futsal training camps

PONE-D-21-37878R2

Dear Dr. Kuo,

We’re pleased to inform you that your manuscript has been judged scientifically suitable for publication and will be formally accepted for publication once it meets all outstanding technical requirements.

Kind regards,

Luca Paolo Ardigò, Ph.D.

Academic Editor

PLOS ONE

Additional Editor Comments (optional):

Congratulations for the interesting work.

Reviewers' comments:

Reviewer's Responses to Questions

**Comments to the Author**

1. If the authors have adequately addressed your comments raised in a previous round of review and you feel that this manuscript is now acceptable for publication, you may indicate that here to bypass the “Comments to the Author” section, enter your conflict of interest statement in the “Confidential to Editor” section, and submit your "Accept" recommendation.

Reviewer #1: All comments have been addressed

2. Is the manuscript technically sound, and do the data support the conclusions?

Reviewer #1: Yes

3. Has the statistical analysis been performed appropriately and rigorously? 

Reviewer #1: Yes

4. Have the authors made all data underlying the findings in their manuscript fully available?

Reviewer #1: No

5. Is the manuscript presented in an intelligible fashion and written in standard English?

Reviewer #1: Yes

6. Review Comments to the Author

Reviewer #1: I would like to congratulate the authors by this revised version of the paper. All ny comments has been taking into account by the authors.

7. PLOS authors have the option to publish the peer review history of their article (what does this mean?). If published, this will include your full peer review and any attached files.

Reviewer #1: No

---

## [Editor Report · Acceptance letter]

8 Apr 2022

PONE-D-21-37878R2 

Relationships between perceived measures of internal load and wellness status during overseas futsal training camps 

Dear Dr. Kuo:

I'm pleased to inform you that your manuscript has been deemed suitable for publication in PLOS ONE. Congratulations! Your manuscript is now with our production department. 

Kind regards, 

on behalf of

Dr. Luca Paolo Ardigò 

Academic Editor

PLOS ONE